# Mean Corpuscular Volume Is Correlated with Liver Fibrosis Defined by Noninvasive Blood Biochemical Indices in Individuals with Metabolic Disorders Aged 60 Years or Older

**DOI:** 10.3390/jcm14134680

**Published:** 2025-07-02

**Authors:** Yousuke Kaneko, Yutaka Kawano, Saki Kawata, Kensuke Mori, Minae Hosoki, Taiki Hori, Kohsuke Miyataka, Seijiro Tsuji, Tomoyo Hara, Hiroki Yamagami, Toshiki Otoda, Tomoyuki Yuasa, Akio Kuroda, Takeshi Harada, Hirokazu Miki, Shingen Nakamura, Itsuro Endo, Munehide Matsuhisa, Ken-ichi Matsuoka, Ken-ichi Aihara

**Affiliations:** 1Department of Internal Medicine, Tokushima Prefectural Kaifu Hospital, 266 Mugi-cho, Tokushima 775-0006, Japanminae@tph.gr.jp (M.H.);; 2Department of Hematology, Endocrinology and Metabolism, Graduate School of Biomedical Sciences, Tokushima University, 3-18-15 Kuramoto-cho, Tokushima 770-8503, Japan; 3Department of Community Medicine and Medical Science, Graduate School of Biomedical Sciences, Tokushima University, 3-18-15 Kuramoto-cho, Tokushima 770-8503, Japan; ykawano@tokushima-u.ac.jp (Y.K.); shingen@tokushima-u.ac.jp (S.N.); 4Department of Diabetes and Metabolism, Tokushima Prefectural Central Hospital, 1-10-3 Kuramoto-cho, Tokushima 770-8539, Japan; 5Department of Endocrinology and Metabolism, Shikoku Medical Center for Children and Adults Hospital, 2-1-1 Senyu-cho, Kagawa 765-8507, Japan; 6Diabetes Therapeutics and Research Center, Institute of Advanced Medical Sciences, Tokushima University, 3-18-15 Kuramoto-cho, Tokushima 770-8503, Japan; 7Division of Transfusion Medicine and Cell Therapy, Tokushima University Hospital, 2-50-1 Kuramoto-cho, Tokushima 770-8503, Japan; 8Department of Bioregulatory Sciences, Graduate School of Biomedical Sciences, Tokushima University, 3-18-15 Kuramoto-cho, Tokushima 770-8503, Japan; 9Department of Internal Medicine, Anan Medical Center, 6-1 Kawahara Takarada-cho, Anan, Tokushima 774-0045, Japan

**Keywords:** MCV, MASLD, liver fibrosis, metabolic disorders, elderlies

## Abstract

**Background:** Metabolic dysfunction-associated steatotic liver disease (MASLD) causes progressive liver fibrosis. Although erythrocyte mean corpuscular volume (MCV) has been shown to have a positive correlation with all-cause mortality, the association between MCV and the development of MASLD has not been fully elucidated. Here, we examined the clinical significance of the association between MCV and MASLD. **Methods:** A cross-sectional study was carried out in 1009 Japanese individuals (including 186 individuals aged < 60 years and 823 individuals aged ≥ 60 years) with metabolic disorders. The relationships between MCV and noninvasive clinical markers of liver fibrosis, including fibrosis-4 (FIB-4) index, aspartate aminotransferase-to-platelet ratio index (APRI), and non-alcoholic fatty liver disease (NAFLD) fibrosis score (NFS), were statistically evaluated. **Results:** Using multiple and logistic regression analyses in overall subjects, it was found that MCV was positively and independently associated with the values of FIB-4 index, APRI, NFS, and the prevalence of liver fibrosis defined by each index. However, the associations between the MCV value and MASLD indices were found to be positive in subjects aged ≥ 60 years but not in those aged < 60 years. **Conclusions:** MCV might be a simple and useful biomarker for the development of MASLD in the elderly.

## 1. Introduction

Metabolic dysfunction-associated steatotic liver disease (MASLD) is a pathological condition that arises from metabolic abnormalities such as obesity and diabetes. Clinical evidence has shown that long-term liver fibrotic changes induced by metabolic disorders can lead to the development of liver cirrhosis and liver cancer [1]. In addition, it has been shown that patients with MASLD who were diagnosed by liver biopsy, even those with simple fatty liver without fibrosis, had a mortality rate that was 1.7 times higher than that in healthy subjects, and the primary cause of death was various cancers including liver cancer [2]. Moreover, it has been reported that MASLD is a crucial risk factor for the incidence and progression of cardiovascular disease (CVD) [3,4] and that MASLD is linked to prognosis in patients with CVD even without known risk factors [5]. Thus, elucidating the pathology of MASLD and establishing a method for the early detection of MASLD are clinical challenges that need to be urgently addressed.

Some noninvasive clinical markers, including fibrosis-4 (FIB-4) index [6,7], aspartate aminotransferase-to-platelet ratio index (APRI) [8,9], and non-alcoholic fatty liver disease (NAFLD) fibrosis score (NFS) [10,11], are now used as alternatives to invasive liver biopsy for estimating liver fibrosis. Although they all are positively related to liver fibrosis in MASLD, they need multiple parameters, and, therefore, a simpler and more convenient score is required for widespread use in clinical practice.

Mean corpuscular volume (MCV) is a clinical marker of the mean size of erythrocytes and dividing the hematocrit (Hct) by red blood cell count gives the value of MCV. MCV is an index that is frequently used in peripheral blood tests and is useful for classifying the cause of anemia. It is known that MCV increases with age. Moreover, it has been reported that MCV shows a positive correlation with all-cause mortality and liver cancer mortality [12].

Hematological abnormalities are frequently present in patients with chronic liver disease. Approximately 75% of patients with chronic liver disease have anemia of diverse etiologies [13]. In patients with severe liver dysfunction, thrombocytopenia, clotting factor deficiency, and various factors cause hemorrhage leading to anemia [14]. In addition, in patients with cirrhosis, especially in patients with alcohol-related cirrhosis, inadequate dietary intake or malabsorption can lead to the deficiency of vitamin B12 and folate, resulting in macrocytic anemia [15]. Thus, MCV shows various values depending on the various pathologies in chronic liver disease.

However, there has been insufficient verification of whether the MCV value reflects the severity of MASLD without severe anemia. This study was, therefore, designed to examine the usefulness of MCV as an indicator for the development of MASLD in individuals with metabolic disorders.

## 2. Materials and Methods

### 2.1. Study Design, Subjects, and Ethics Statement

This study was a retrospective, cross-sectional, and multicenter study. All of the subjects were older than 20 years of age and were enrolled from the Department of Internal Medicine in Tokushima Prefectural Kaifu Hospital, Department of Endocrinology and Metabolism in Tokushima University Hospital, Department of Internal Medicine in Anan Medical Center, Department of Internal Medicine in Shikoku Medical Center for Children and Adults, and Department of Internal Medicine in Takamatsu Municipal Hospital (all of the medical institutes were located in Shikoku Island, Japan) between January 2018 and March 2024.

We obtained information from electronic medical records on clinical characteristics of 1009 Japanese individuals (555 men and 454 women) who were outpatients or inpatients with at least one metabolic disorder including obesity, type 2 diabetes mellitus (T2DM), hypertension, and dyslipidemia.

T2DM was diagnosed in accordance with the criteria issued by the Expert Committee on the Diagnosis and Classification of Diabetes Mellitus [16]. Current smokers were defined as those who had smoked within the last two years. The calculation of body mass index (BMI) was used as an indicator of obesity. An automatic sphygmomanometer was used to measure blood pressure (BP) in each participant while sitting. A diagnosis of hypertension was defined as showing systolic blood pressure (SBP) ≥ 140 mmHg and/or showing diastolic blood pressure (DBP) ≥ 90 mmHg, or taking antihypertensive medication. A diagnosis of dyslipidemia was defined as manifesting low-density lipoprotein cholesterol (LDL-C) ≥ 140 mg/dL, fasting triglyceride (TG) ≥ 150 mg/dL or casual TG ≥ 175 mg/dL, high-density lipoprotein cholesterol (HDL-C) < 40 mg/dL, or taking lipid-lowering medication. The following patients with metabolic disorders were excluded from this study: (1) patients with advanced cancer; (2) patients who were pregnant; (3) patients who consumed ≥140 g and ≥210 g of alcohol per week for females and males, respectively; (4) patients with liver disease other than MASLD; (5) patients with progressed kidney disease who exhibited a serum creatinine (Cr) level higher than 2.0 mg/dL; (6) patients who had an Hct level lower than 30% or a hemoglobin (Hgb) level lower than 10.0 g/dL; (7) patients receiving blood transfusions and patients treated with hematopoietic agents such as iron, folic acid, mecobalamin, erythropoietin stimulating factor, and hypoxia-inducible factor prolyl hydroxylase inhibitors.

The electronic medical records of each hospital were utilized to obtain information about subjects’ clinical characteristics and laboratory data, and all data were anonymized. Because this clinical study was a retrospective observational study, informed consent was not required from individual subjects; instead, a notice was posted at each hospital regarding the option of not participating in the study.

The implementation plan for this retrospective study, which did not require informed consent, was approved by the Institutional Review Board, Ethics Committee of Anan Medical Center, the primary medical institution for this clinical study, and was conducted with the comprehensive approval of all directors of the affiliated hospitals (Tokushima Prefectural Kaifu Hospital, Tokushima University Hospital, Shikoku Medical Center for Children and Adults, and Takamatsu Municipal Hospital (approval ID: 202315, approval date: 11 March 2024); Ethics Committee of Anan Medical Center). This clinical study was conducted in accordance with the Declaration of Helsinki and “Ethical Guidelines for Medical and Biological Research Involving Human Subjects” issued by the Ministry of Education, Culture, Sports, Science and Technology Japan (accessed on 14 April 2025, https://www.mext.go.jp/content/20250325-mxt_life-000035486-01.pdf).

### 2.2. Biochemical Analysis

Peripheral blood test for blood cell counts with MCV determination was carried out with automatic blood cell analyzer. Biochemical parameters including plasma glucose (PG), hemoglobin A1c (HbA1c), LDL-C, TG, HDL-C, uric acid (UA), aspartame aminotransferase (AST), alumni aminotransferase (ALT), creatinine (Cr), and albumin (ALB) were measured by the enzymatic method and high-performance liquid chromatography.

### 2.3. Determination of Severity of Steatotic Liver Disease Represented by HSI, FIB-4 Index, APRI, and NFS

Hepatic steatosis index (HSI) as a clinical marker of fatty liver was calculated in each subject by using the following formula: 8× (ALT/AST ratio) + BMI (+2 if female; +2 if diabetes mellitus is present). The presence of a fatty liver is suggested when it is greater than 36 [17]. We also used three clinical scoring systems, including fibrosis-4 (FIB-4) index, aspartate aminotransferase-to-platelet ratio index (APRI), and non-alcoholic fatty liver disease (NAFLD) fibrosis score (NFS), for the evaluation of liver fibrosis. To measure the severity of MASLD, each index was determined according to the following formulas.

*FIB-4 index = Age (years) × AST (U/L)/[platelet count (×10^9^/L) × √ALT (U/L)] [6]*APRI = 100 × AST (U/L)/upper limit of normal AST (U/L)/platelet count (×10^9^/L) [8]*NFS = −1.675 + 0.037 × age (years) + 0.094 × BMI (kg/m^2^) + 1.13 × IFG/diabetes (yes = 1, no = 0) + 0.99 × AST/ALT ratio − 0.013 × platelet count (× 10^9^/L) − 0.66 × ALB (g/dL) [10]

Cutoff points of the clinical score were determined according to the following criteria: FIB-4 index < 1.3 was classified as low risk of liver fibrosis, FIB-4 index ≥ 1.3 and <2.67 were classified as intermediate risk of liver fibrosis, and FIB-4 index > 2.67 was classified as high risk of developed liver fibrosis. APRI >0.5 was considered to have liver fibrosis, and APRI > 1.5 was considered to have cirrhosis. NFS < −1.455 was the criterion for no liver fibrosis, NFS −1.455 to 0.676 was the criterion for intermediate liver fibrosis, and NFS ≥ 0.676 was the criterion for advanced liver fibrosis.

The correlations between the severity of MASLD and clinical confounding factors would differ in elderly individuals and non-elderly individuals due to the inclusion of age in the formulas for the FIB-4 index and NFS. Therefore, in addition to the overall analysis, we conducted analyses for subjects divided into a group of subjects who were under 60 years of age and a group of subjects who were 60 years of age or older.

### 2.4. Statistical Analysis

Using the Shapiro–Wilk test, the normality of the distribution of continuous variables was determined. Means and standard deviation (SD) were used to express continuous variables with a normal distribution, while medians (Q1, Q3) were used to express those with a non-normal distribution. Numbers or percentages were used to express categorical parameters. Dummy variables were assigned to males; subjects who had diabetes, hypertension, or dyslipidemia; and subjects who were current smokers. Multivariate regression and logistic regression analyses were used to determine the relationships between clinical markers of fatty liver (HSI) or liver fibrosis (FIB-4 index, APRI, and NFS) and clinical variables including MCV. Moreover, a receiver operating characteristic (ROC) curve analysis was used to determine the optimal cutoff value of MCV for identifying advanced stages of liver fibrosis. All of the analyses were conducted using GraphPad Prism 10 (GraphPad Software, San Diego, CA, USA). The threshold for statistical significance was set as *p* < 0.05.

## 3. Results

### 3.1. Characteristics of the Subjects

The physical and laboratory characteristics of the participants in the study are shown in Table 1. On average, HbA1c, estimated glomerular filtration rate (eGFR), RBC, Hct, Hgb, platelets counts, and serum levels of LDL-C, TG, ALB, AST, and ALT were higher in subjects aged < 60 years than in subjects aged ≥ 60 years. MCV and serum levels of HDL-C and Cr were higher in subjects aged ≥ 60 years than in subjects aged < 60 years. The percentages of current smokers and subjects with diabetes were higher in subjects aged < 60 years than in subjects aged ≥ 60 years. The prevalence of hypertension was higher in subjects aged ≥ 60 years than in subjects aged < 60 years. The percentages of subjects using metformin, insulin, and glucagon-like peptide-1 receptor agonist (GLP-1RA) were higher in subjects aged < 60 years than in subjects aged ≥ 60 years. The percentages of subjects using angiotensin II receptor blocker (ARB)/angiotensin-converting enzyme inhibitor (ACEi), calcium channel blocker (CCB), statin, antiplatelet, sulfonylurea (SU)/glinide, and dipeptidyl peptidase-4 inhibitor (DPP-4i) were higher in subjects aged ≥ 60 years than in subjects aged < 60 years. Subjects aged < 60 years had a higher level of HSI than that in subjects aged ≥ 60 years. Subjects aged ≥ 60 years had a higher level of FIB4-index and a higher NFS than those in subjects aged < 60 years.

### 3.2. Associations Between MCV and Noninvasive MASLD Indices

Scatter plots showed that MCV had a negative correlation with HSI in the overall subjects (Figure 1a). MCV also had a negative correlation with HSI in the two groups of subjects aged <60 years (*p* = 0.001) and subjects aged ≥60 years (*p* < 0.001) (Figure 1b,c). However, multiple regression analysis showed that MCV did not have any significant correlation with HSI (total subjects: *p* = 0.327, subjects aged <60 years: *p* = 0.876, and subjects aged ≥60 years: *p* = 0.401) (Table 2, Table 3 and Table 4).

On the other hand, in the overall subjects, MCV had positive correlations with FIB- 4 index (*p* < 0.001), APRI (*p* < 0.001), and NFS (*p* < 0.001) as shown in Figure 1a. Multiple regression analysis confirmed that MCV was an independent and the sole common clinical factor having positive correlations with the liver fibrosis indices, including FIB-4 index (*p* < 0.001), APRI (*p* < 0.001), and NFS (*p* = 0.002) (Table 2). In the age group-specific analyses, MCV had positive correlations with the liver fibrosis indices (FIB-4 index (*p* < 0.001), APRI (*p* < 0.001), and NFS (*p* = 0.004)) in the subjects aged ≥60 years (Table 4) but not in the subjects aged <60 years (Table 3).

To evaluate the effects of medication and finally determine the factors that influence liver fibrosis, we performed multiple regression analyses using clinical variables with significant correlations as shown in Table 2, Table 3 and Table 4 and medications taken by the subjects. As a result, MCV showed significant and positive correlations with the liver fibrosis indices in the overall subjects regardless of the medications taken (FIB-4 index: *p* < 0.001, APRI: *p* < 0.001, NFS: *p* = 0.002) (Appendix A). Furthermore, positive correlations between MCV and the three clinical indices of liver fibrosis (FIB-4 index (*p* < 0.001), APRI (*p* < 0.001), and NFS (*p* = 0.003)) were found in subjects aged ≥60 years but not in subjects aged <60 years after the adjustment of medications used (Appendix A).

### 3.3. Association Between MCV and Prevalence of Advanced Liver Fibrosis

Next, we performed logistic regression analyses with confounding factors that were identified by multiple regression analyses for the prevalence of liver fibrosis based on the cutoff value of each MASLD index. In the present study, advanced liver fibrosis was defined as the FIB-4 index exceeding 2.67 [6], APRI exceeding 0.5 [8], or NFS exceeding 0.676 [10]. As shown in Figure 2a, MCV is the only common factor that shows positive correlations with the clinical indices of advanced liver fibrosis in the overall subjects (FIB-4 index: increase in odds ratio with MCV at 1fL to 1.051, *p* = 0.014; APRI: increase in odds ratio with MCV at 1fL to 1.070, *p* < 0.001; NFS: increase in odds ratio with MCV at 1fL to 1.082, *p* < 0.001). In the analysis of subjects aged ≥60 years, MCV also showed positive correlations with the three clinical markers (FIB-4 index: increase in odds ratio with MCV at 1fL to 1.056, *p* = 0.009; APRI: increase in odds ratio with MCV at 1fL to 1.054, *p* = 0.003; NFS: increase in odds ratio with MCV at 1fL to 1.084, *p* < 0.001) (Figure 2b).

We performed ROC analysis to determine the cutoff value of MCV indicating advanced liver fibrosis for each clinical marker. Finally, the cutoff values of MCV for the determination of advanced liver fibrosis in the overall subjects were ≥95.55, ≥94.65, and ≥93.65 for the FIB-4 index, APRI, and NFS, respectively (Figure 3a). In the analysis of the subjects ≥60 years, the cutoff values of MCV for the determination of advanced liver fibrosis were ≥95.55, ≥94.65, and ≥93.75 for the FIB-4 index, APRI, and NFS, respectively (Figure 3b).

## 4. Discussion

In chronic liver disease, macrocytic changes occur through various mechanisms. Folate or vitamin B12 deficiency causes ineffective or dysplastic erythropoiesis due to impaired DNA synthesis. This results in the production of erythroblasts, which undergo macrocytic transformation [18]. In addition, it has been reported that vitamin B12 levels are lower in MASLD patients than in healthy individuals, and there is a negative correlation between vitamin B12 levels and the severity of liver steatosis [19]. Therefore, it seems that vitamin B12 is crucial for maintaining the homeostasis of both the red blood cell and liver metabolism.

Since immature erythrocytes are larger than mature erythrocytes, increasing reticulocytes due to hemorrhage or hemolysis causes macrocytic change in patients with severe liver dysfunction such as cirrhosis [15]. Furthermore, the membrane lipid composition of mature red blood cells is maintained by equilibrium with blood lipids [20]. In cirrhosis with advanced liver fibrosis, reflecting lipid metabolism abnormalities, free cholesterol in the red blood cell membrane increases, causing macrocytic change [21,22].

In MASLD, an increase in free fatty acids and excess lipids causes an increase in active oxygen and a decline in the antioxidant system in the liver, leading to the progression of liver fibrosis [23,24]. In primary biliary cirrhosis, oxidative stress and nitrosative stress are associated with ongoing manifestation of the disease [25]. Furthermore, a relationship between increased oxidative stress and the progression of MASLD has been shown in both clinical studies and experimental animal studies [26,27,28]. On the other hand, it has been demonstrated that augmented oxidative stress induced by irradiation and anticancer chemotherapy accelerates hemolysis and eryptosis with increasing MCV value and membrane blebbing through Ca^2+^ accumulation and the activation of RAC-1 GTPase of erythrocytes. Therefore, increased MCV may serve as a biomarker of increased oxidative stress in vivo, which may lead to the development of MASLD [29,30,31].

Chronic inflammation contributes to the pathology of various chronic diseases. Among them, psoriasis is well known as a chronic, recurrent, inflammatory skin disease. It is well known that inflammatory cytokines are elevated in both tissues and peripheral circulation in patients with psoriasis [32]. Since it has been reported that patients with psoriasis have higher MCV and C-reactive protein (CRP) levels compared to healthy subjects [33], MCV may be related to chronic inflammation that is crucial for the development of liver fibrosis in MASLD.

Clonal hematopoiesis of indeterminate potential (CHIP) is considered to be a myeloid precursor lesion. Individuals with CHIP have been shown to have increased risks of developing myeloid malignancies such as leukemia and myelodysplastic syndrome (MDS), CVDs, and various inflammatory diseases [34]. CHIP was shown to be associated with an increased risk of MASLD via accelerated inflammatory responses by Wong and colleagues who conducted a combination of large-scale human genetic studies and an animal study using hematopoietic-specific *Tet2*-deficient mice as a clonal hematopoiesis model manifesting severe steatohepatitis with increased liver inflammation and fibrosis [35]. Furthermore, they revealed that CHIP is a modifiable risk factor for MASLD by enhancing NLR family pyrin domain-containing protein 3 (NLRP3) inflammasome and its downstream inflammatory cytokines [35]. Since an increase in MCV is a characteristic of MDS, which is also known as CHIP-related disease, it is possible that an increase in MCV reflects an increase in CHIP, causing an activation of NLRP3 inflammasome and inflammatory cytokines that results in an inflammatory condition in the liver [35]. In support of this hypothesis, MCV had positive correlations with liver fibrosis indices in patients aged ≥60 years but not in patients aged <60 years in the present study. In a previous study, CHIP was identified in less than 1% of individuals under 50 years of age, while it was identified in more than 10% of those older than 65 years of age [36]. Therefore, CHIP may be involved in the increase in MCV in elderly patients with MASLD.

In this study, metformin showed negative correlations with FIB-4 index and APRI. Previous studies suggested that AMPK activators, including metformin, can improve insulin resistance, liver steatosis, and dyslipidemia through the inactivation of sterol regulatory element-binding protein (SREBP) [37], hepatic stellate cell activation [38,39], and the TGF-β1/Smad3 pathway [40]. In addition, a clinical study showed that metformin reduces intrahepatic fat as assessed by ultrasonography in patients with MASLD [41]. The results of those studies support the results of our clinical study showing a hepatoprotective effect of metformin against the development of MASLD.

## 5. Limitations

The present study has several limitations as follows. The findings of this study are limited to individuals with metabolic disorders, and the findings, therefore, cannot be extended to the general population. Second, imaging examinations including elastography using ultrasound or MRI for the assessment of liver fibrosis and stiffness were not performed in this study. Confirming our hypothesis would be possible by analyzing the association between the MCV value and severity of liver stiffness evaluated by those noninvasive imaging methods. Third, in cases in which a liver biopsy is required for the pathological evaluation of MASLD, the relationship between histopathological findings and MCV needs to be evaluated. Forth, it is unclear whether normalizing the enlarged MCV with nutritional or pharmaceutical interventions that reduce oxidative stress can improve the severity of MASLD. This warrants investigation in future prospective studies. Finally, the absence of a causal relationship between MCV and the development of MASLD in this cross-sectional study requires additional large-scale and longitudinal analyses with additional biomarkers for the assessment of inflammation, oxidative stress, and CHIP to clarify the clinical question.

## 6. Conclusions

The present study revealed that MCV had an independent and positive correlation with the development of liver fibrosis in individuals aged ≥60 years with metabolic disorders. Thus, elderly patients who have a higher MCV level (above 95 fL) may require additional evaluation for the development of liver fibrosis and prompt comprehensive intervention for metabolic disorders. This study’s findings and predicted mechanism are presented in Figure 4.

## Figures and Tables

**Figure 1 jcm-14-04680-f001:**
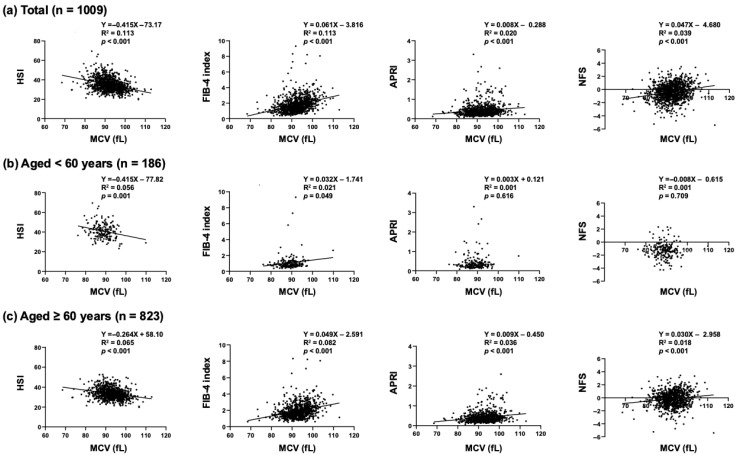
Scatter plots between MCV and MASLD indices in (**a**) overall subjects, (**b**) subjects aged < 60 years, and (**c**) subjects aged ≥ 60 years.

**Figure 2 jcm-14-04680-f002:**
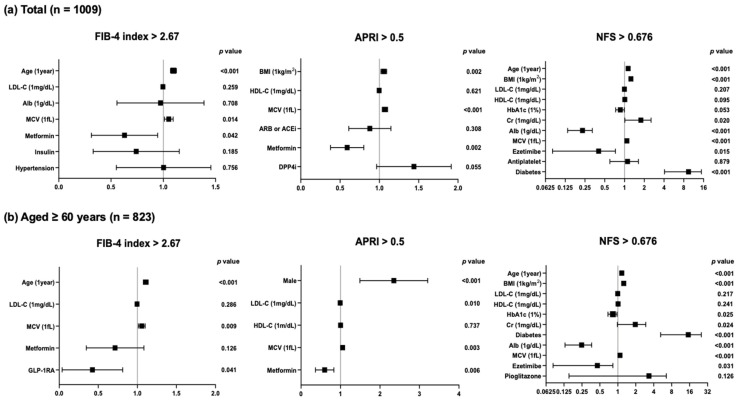
Logistic regression analyses with confounding factors identified by multiple regression analyses for the prevalence of liver fibrosis based on MASLD indices in (**a**) overall subjects and (**b**) subjects aged ≥60 years.

**Figure 3 jcm-14-04680-f003:**
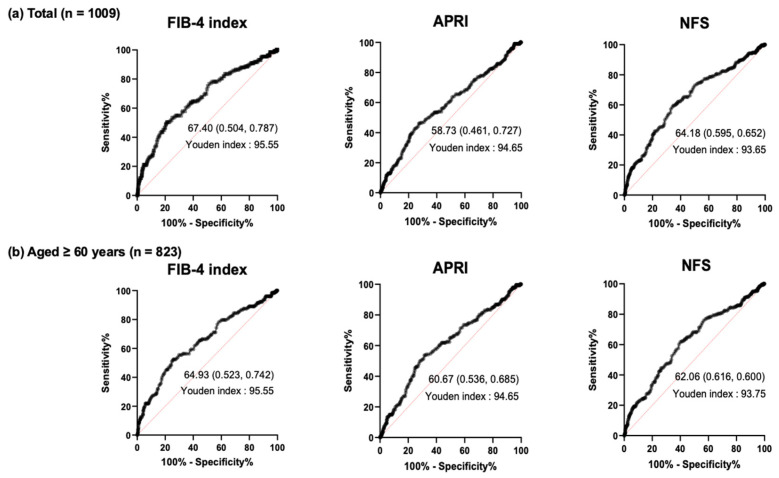
ROC curve analysis of MCV value for determination of liver fibrosis based on MASLD indices in (**a**) overall subjects and (**b**) subjects aged ≥60 years.

**Figure 4 jcm-14-04680-f004:**
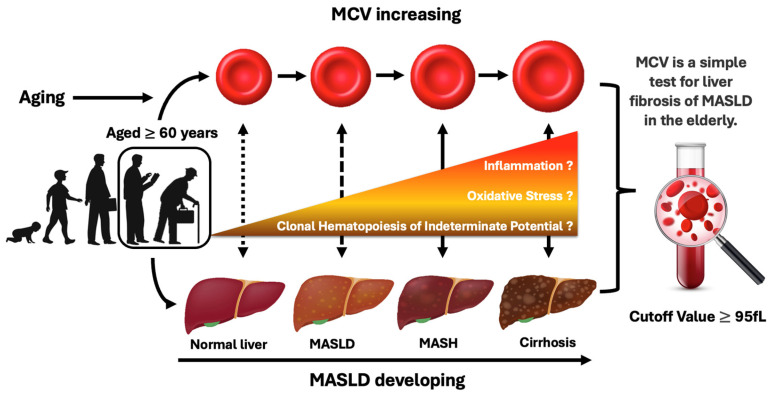
Summary illustration of the relationship between the increased MCV of erythrocytes and the progression of MASLD in elderly people with metabolic disorders.

**Table 1 jcm-14-04680-t001:** Clinical characteristics of the subjects.

Groups/Clinical Factors	Total (*n* = 1009)	<60 Years (*n* = 186)	≥60 Years (*n* = 823)	*p* Value (<60 vs. ≥60)
Males/Females	555/454	108/78	447/376	0.288
Age (years)	72.0 (63.0, 78.0)	52.7 (46.0, 56.0)	74.0 (68.8, 80.0)	<0.001
BMI (kg/m^2^)	24.2 (21.7, 27.0)	27.1 (24.2, 30.8)	23.7 (21.4, 26.2)	<0.001
SBP (mmHg)	133.6 ± 16.9	131.9 ± 16.8	134.0 ± 16.9	0.127
LDL-C (mg/dL)	100 (81, 122)	111.0 (83.3, 134.0)	98 (81, 119)	<0.001
TG (mg/dL)	116 (81, 161)	136.0(99.3, 202.8)	110.0 (78.5, 155.5)	<0.001
HDL-C (mg/dL)	54 (45, 66)	51.0 (42.0, 60.8)	55 (46, 67)	<0.001
Casual PG (mg/dL)	129 (109, 171)	137.0 (110.2, 186.5)	128 (108, 167)	0.077
HbA1c (%)	6.6 (6.1, 7.4)	7.0 (6.3, 7.8)	6.6 (6.0, 7.3)	<0.001
UA (mg/dL)	5.1 (4.2, 6.1)	5.4 (4.2, 6.2)	5.0 (4.2, 6.0)	0.153
Cr (mg/dL)	0.8 (0.65, 0.97)	0.72 (0.60, 0.87)	0.82 (0.67, 1.01)	<0.001
eGFR (mL/min/1.73 m^2^)	67.0 ± 22.0	82.3 ± 22.6	63.6 ± 20.3	<0.001
ALB (g/dL)	4.1 ± 0.5	4.3 ± 0.5	4.1 ± 0.5	<0.001
AST (U/L)	21 (17, 26)	21 (17, 28)	21 (17, 26)	0.031
ALT (U/L)	19 (14, 28)	26 (18.0, 43.8)	18 (13, 26)	<0.001
RBC (10^12^/L)	4.56 (4.14, 4.94)	5.00 (4.64, 5.28)	4.47 (4.07, 4.83)	<0.001
Hct (%)	41.7 (38.5, 44.9)	44.2 (40.6, 46.9)	41.4 (38.0, 44.3)	<0.001
Hgb (g/dL)	13.8 (12.6 14.9)	14.9 (13.5, 15.9)	13.7 (12.4, 14.6)	<0.001
MCV (fL)	92.0 (88.9, 95.6)	89.1 (86.0, 91.6)	92.9 (89.6, 96.5)	<0.001
Platelets (10^9^/L)	215 (182, 255)	239.5 (203.5, 277.0)	210.0 (178.0, 249.5)	<0.001
HSI	34.0 (30.5, 38.4)	40.1 (35.6, 45.3)	33.1 (29.8, 36.9)	<0.001
FIB-4 index	1.58 (1.14, 2.17)	0.85 (0.68, 1.19)	1.77 (1.33, 2.31)	<0.001
APRI	0.35 (0.26, 0.47)	0.32 (0.23, 0.44)	0.36 (0.27, 0.47)	0.732
NFS	−0.35 (−1.15, 0.51)	−1.28 (−2.02, −0.66)	−0.14 (−0.88, 0.64)	<0.001
Current Smoker (*n*, (%))	108 (10.7)	38 (20.4)	70 (8.5)	<0.001
Hypertension (*n*, (%))	719 (71.3)	109 (58.6)	610 (74.1)	<0.001
Dyslipidemia (*n*, (%))	745 (73.8)	142 (76.3)	603 (73.3)	0.389
Diabetes (*n*, (%))	770 (76.3)	162 (87.1)	608 (73.9)	<0.001
ARB or ACEi (*n*, (%))	473 (46.9)	64 (34.4)	409 (49.7)	<0.001
CCB (*n*, (%))	440 (43.6)	60 (32.3)	380 (46.2)	0.001
β blocker (*n*, (%))	103 (10.2)	16 (8.6)	87 (10.6)	0.423
MR antagonist (*n*, (%))	40 (4.0)	3 (1.6)	37 (4.5)	0.069
Statin (*n*, (%))	515 (51.0)	73 (39.2)	442 (53.7)	<0.001
Ezetimibe (*n*, (%))	71 (7.0)	13 (7.0)	58 (7.0)	0.978
Antiplatelet (*n*, (%))	172 (17.0)	14 (7.5)	158 (19.2)	<0.001
SU or Glinide (*n*, (%))	139 (13.8)	16 (8.6)	123 (14.9)	0.023
Metformin (*n*, (%))	364 (36.1)	86 (46.2)	278 (33.8)	0.001
DPP-4i (*n*, (%))	430 (42.6)	62 (33.3)	368 (44.7)	0.005
SGLT2i (*n*, (%))	333 (33.0)	68 (36.6)	265 (32.2)	0.253
α-GI (*n*, (%))	94 (9.3)	16 (8.6)	78 (9.5)	0.711
Pioglitazone (*n*, (%))	22 (2.2)	3 (1.6)	19 (2.3)	0.557
Insulin (*n*, (%))	192 (19.0)	47 (25.3)	145 (17.6)	0.016
GLP-1RA (*n*, (%))	116 (11.5)	43 (23.1)	73 (8.9)	<0.001

Values are presented as means ± SD or medians (Q1, Q3). Abbreviations: BMI: body mass index; SBP: systolic blood pressure; LDL-C: low-density lipoprotein cholesterol; TG: triglycerides; HDL-C: high-density lipoprotein cholesterol; PG: plasma glucose; HbA1c: hemoglobin A1c; UA: uric acid; Cr: creatinine; eGFR: estimated glomerular filtration rate; ALB: albumin; AST: aspartate aminotransferase; ALT: alanine aminotransferase; RBC: red blood cell; Hct: hematocrit; Hgb: hemoglobin; MCV: mean corpuscular volume; HSI: hepatic steatosis index; FIB-4: fibrosis-4; APRI: aspartate aminotransferase-to-platelet ratio index; NFS: non-alcoholic fatty liver disease fibrosis score; ARB: angiotensin II receptor blocker; ACEi: angiotensin-converting enzyme inhibitor; CCB: calcium channel blocker; MR: mineral corticoid receptor; SU: sulfonylurea; DPP-4i: dipeptidyl peptidase-4 inhibitor; SGLT2i: sodium–glucose cotransporter 2 inhibitor; αGI: alpha-glucosidase inhibitor; GLP-1RA: glucagon-like peptide-1 receptor agonist.

**Table 2 jcm-14-04680-t002:** Multiple regression analysis for determinants of MASLD indices in overall subjects.

		HSI		FIB-4 Index		APRI		NFS	
Variables	VIF	*t* Value	*p* Value	*t* Value	*p* Value	*t* Value	*p* Value	*t* Value	*p* Value
Age	1.747	−8.88	<0.001	11.63	<0.001	0.01	0.991	18.65	<0.001
Male	1.480	−6.46	<0.001	0.51	0.608	1.95	0.052	0.60	0.546
Current Smoking	1.150	0.01	0.995	0.21	0.832	0.02	0.981	−0.49	0.624
Hypertension	1.262	0.05	0.957	−1.99	0.047	−1.11	0.267	−1.63	0.104
Dyslipidemia	1.133	0.80	0.423	−0.97	0.334	−0.31	0.756	−1.33	0.183
Diabetes	1.426	10.49	<0.001	−0.60	0.548	1.09	0.276	13.51	<0.001
BMI	1.493	52.77	<0.001	0.63	0.528	2.42	0.016	11.69	<0.001
SBP	1.199	−1.04	0.301	0.78	0.433	−0.02	0.985	0.84	0.400
LDL-C	1.181	1.28	0.199	−2.80	0.005	−1.92	0.055	−2.69	0.007
TG	1.355	1.12	0.263	−0.72	0.469	0.02	0.982	−0.78	0.434
HDL-C	1.452	−1.60	0.110	1.80	0.073	2.91	0.004	2.25	0.024
HbA1c	1.409	2.53	0.012	−1.44	0.152	0.04	0.967	−2.52	0.012
UA	1.397	−0.70	0.482	1.69	0.079	1.55	0.122	0.89	0.373
Cr	1.422	−2.19	0.029	1.76	0.113	−0.10	0.920	2.49	0.013
ALB	1.360	4.61	<0.001	−2.36	0.018	−0.45	0.651	−10.13	<0.001
Hct	1.509	2.17	0.031	−1.03	0.301	−0.21	0.835	0.49	0.626
MCV	1.367	−0.98	0.327	3.85	<0.001	3.78	<0.001	3.06	0.002

VIF: variance inflation factor.

**Table 3 jcm-14-04680-t003:** Multiple regression analysis for determinants of MASLD indices in subjects aged < 60 years.

		HSI		FIB-4 Index		APRI		NFS	
Variables	VIF	*t* Value	*p* Value	*t* Value	*p* Value	*t* Value	*p* Value	*t* Value	*p* Value
Age	1.359	−2.15	0.033	3.67	<0.001	0.97	0.333	5.08	<0.001
Male	1.749	−0.66	0.513	−0.43	0.667	0.13	0.897	−0.30	0.765
Current Smoking	1.257	0.18	0.860	−0.27	0.791	−0.84	0.404	0.54	0.589
Hypertension	1.705	−0.19	0.847	−2.34	0.020	−1.83	0.069	−0.69	0.490
Dyslipidemia	1.229	1.05	0.295	−1.39	0.166	−0.71	0.479	−1.06	0.293
Diabetes	1.358	1.82	0.071	0.68	0.498	0.58	0.560	4.47	<0.001
BMI	1.573	21.55	<0.001	0.91	0.366	1.56	0.121	6.41	<0.001
SBP	1.521	0.47	0.642	0.59	0.559	−0.25	0.799	0.07	0.944
LDL-C	1.464	0.24	0.810	−1.05	0.295	−0.40	0.692	−2.19	0.030
TG	1.412	−0.26	0.794	−0.10	0.921	0.42	0.672	0.27	0.785
HDL-C	1.753	−0.92	0.360	1.07	0.286	1.13	0.260	1.24	0.219
HbA1c	1.551	0.42	0.675	−0.26	0.799	−0.11	0.910	−0.27	0.790
UA	1.407	−0.66	0.513	1.18	0.240	0.44	0.664	1.15	0.253
Cr	1.235	−0.27	0.788	0.44	0.658	0.13	0.899	0.22	0.824
ALB	1.368	2.77	0.006	−1.17	0.245	−0.18	0.858	−3.05	0.003
Hct	1.486	0.45	0.653	−1.00	0.319	0.18	0.860	0.34	0.731
MCV	1.290	−0.16	0.876	1.25	0.212	0.54	0.588	0.21	0.832

**Table 4 jcm-14-04680-t004:** Multiple regression analysis for determinants of MASLD indices in subjects aged ≥ 60 years.

		HSI		FIB-4 Index		APRI		NFS	
Variables	VIF	*t* Value	*p* Value	*t* Value	*p* Value	*t* Value	*p* Value	*t* Value	*p* Value
Age	1.487	−5.60	<0.001	9.12	<0.001	0.75	0.454	12.3	<0.001
Male	1.451	−7.08	<0.001	0.65	0.519	2.29	0.022	0.59	0.553
Current Smoking	1.148	−0.05	0.958	0.48	0.633	0.77	0.440	−0.47	0.640
Hypertension	1.207	0.00	0.999	−0.56	0.574	0.42	0.672	−1.15	0.251
Dyslipidemia	1.162	0.39	0.693	−0.25	0.801	0.25	0.806	−0.67	0.502
Diabetes	1.451	11.08	<0.001	−0.61	0.543	1.20	0.232	12.83	<0.001
BMI	1.300	46.79	<0.001	−0.41	0.685	1.27	0.203	8.92	<0.001
SBP	1.174	−1.68	0.094	1.03	0.302	0.53	0.599	1.02	0.307
LDL-C	1.145	1.28	0.200	−2.88	0.004	−2.35	0.019	−2.06	0.040
TG	1.326	1.76	0.078	−0.98	0.329	−0.34	0.734	−1.05	0.293
HDL-C	1.429	−0.76	0.448	1.46	0.144	2.66	0.008	2.00	0.046
HbA1c	1.382	3.12	0.002	−1.90	0.058	−0.28	0.779	−2.86	0.004
UA	1.454	−0.36	0.718	1.01	0.314	1.25	0.210	0.18	0.855
Cr	1.486	−2.68	0.007	1.75	0.080	−0.13	0.893	2.83	0.005
ALB	1.361	3.61	<0.001	−1.79	0.073	−0.45	0.653	−9.57	<0.001
Hct	1.539	1.87	0.062	0.08	0.936	0.10	0.918	0.75	0.452
MCV	1.282	−0.84	0.401	3.58	<0.001	4.23	<0.001	2.93	0.004

## Data Availability

The data supporting the findings of this study are available on request from the corresponding author.

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
