# Peer review of "Mean Corpuscular Volume Is Correlated with Liver Fibrosis Defined by Noninvasive Blood Biochemical Indices in Individuals with Metabolic Disorders Aged 60 Years or Older"

_jcm, 2025, doi:10.3390/jcm14134680_

Round 1
Reviewer 1 Report
Comments and Suggestions for Authors
Kaneko et al. reported that MCV significantly and independently contributed to the increases in FIB-4 index, APRI, NFS and the prevalence of liver fibrosis defined by each index. However, the associations between MCV value and MASLD. The manuscript seems well written.
- Although title is “Mean Corpuscular Volume is Correlated with Liver Fibrosis in Individuals with Metabolic Disorders Aged 60 Years or Older”, make change of title: Mean Corpuscular Volume May Be Correlated with Liver Fibrosis in Individuals with Metabolic Disorders Aged 60 Years or Older. Because authors examined only Fib-4 and APRI.
Author Response
(Reviewer’s Comments)
Kaneko et al. reported that MCV significantly and independently contributed to the increases in FIB-4 index, APRI, NFS and the prevalence of liver fibrosis defined by each index. However, the associations between MCV value and MASLD. The manuscript seems well written.
- Although title is “Mean Corpuscular Volume is Correlated with Liver Fibrosis in Individuals with Metabolic Disorders Aged 60 Years or Older”, make change of title: Mean Corpuscular Volume May Be Correlated with Liver Fibrosis in Individuals with Metabolic Disorders Aged 60 Years or Older. Because authors examined only Fib-4 and APRI
(Response)
Thank you very much for taking the time to review this manuscript.
The non-invasive liver fibrosis indices used in this study, such as the FIB-4 index, NFS, and APRI, have been widely recognized as useful in previous clinical studies. In addition, we believe that the accuracy of the study was sufficiently ensured by using three liver fibrosis indices at once in this study. In addition, the results of multivariate analysis clearly demonstrated that there is an independence in the correlation between MCV and these indices. However, since this study did not perform imaging diagnosis, liver stiffness tests, or biopsies, we fully understand the reviewer's concerns. Therefore, we have revised the title of this manuscript as follows. “Mean Corpuscular Volume is Correlated with Liver Fibrosis Defined by Non-invasive Blood Biochemical Indices in Individuals with Metabolic Disorders Aged 60 Years or Older”
Reviewer 2 Report
Comments and Suggestions for Authors
Elevated MCV (macrocytosis) is frequently observed in individuals with liver diseases. This is primarily due to changes in the membrane lipid composition of red blood cells, leading to an increase in surface area and volume. The severity of liver damage can be associated with the degree of macrocytosis, and high values of MCV are correlated to high liver cancer risk. This article examined correlation between MCV and liver fibrosis parameters. Authors showed that high MCV can be a sign that the patient is more in risk of liver fibrosis development. In abstract authors should mention that the correlation is positive or negative.
There are lot of limitations in this study, and still missing molecular connections to explain these correlations. So this may be a very first study that is going in this direction, however, there is a need for more research to confirm this.
In discussion authors could try to explain all potential mechanisms that lead to MCV increase in liver fibrosis such as increased membrane lipids, increased cholesterol, vitamin deficiencies, hepatocyte inflammation.
it would be useful if there is one table or graph or diagram that summarize the major findings.
Author Response
(Reviewer’s Comments)
- Elevated MCV (macrocytosis) is frequently observed in individuals with liver diseases. This is primarily due to changes in the membrane lipid composition of red blood cells, leading to an increase in surface area and volume. The severity of liver damage can be associated with the degree of macrocytosis, and high values of MCV are correlated to high liver cancer risk. This article examined correlation between MCV and liver fibrosis parameters. Authors showed that high MCV can be a sign that the patient is more in risk of liver fibrosis development. In abstract authors should mention that the correlation is positive or negative.
(Response)
Thank you for your valuable comments. In accordance with the reviewer’s suggestion, we revised the results of abstract as follows: Results: Using multiple and logistic regression analyses in overall subjects, it was found that MCV positively and independently associated with the values of FIB-4 index, APRI, NFS and the prevalence of liver fibrosis defined by each index.
- There are lot of limitations in this study, and still missing molecular connections to explain these correlations. So this may be a very first study that is going in this direction, however, there is a need for more research to confirm this. In discussion authors could try to explain all potential mechanisms that lead to MCV increase in liver fibrosis such as increased membrane lipids, increased cholesterol, vitamin deficiencies, hepatocyte inflammation.
(Response)
We thank the reviewer for your thoughtful suggestions. We revised the discussion section and added additional descriptions and references about potential mechanisms that lead to greater MCV and liver fibrosis (lines 286 to 289, lines 293 to 296 and lines 308 to 314).
- it would be useful if there is one table or graph or diagram that summarize the major findings.
(Response)
We thank the reviewers for your very helpful advice. To enable the results of this study and the predicted pathological mechanisms to be understood at a glance, a graphical abstract has been included in the main text as figure 4.
Reviewer 3 Report
Comments and Suggestions for Authors Investigate the biological pathways connecting MCV variations to liver fibrosis, including oxidative stress indicators, CHIP-associated genetic modifications, and inflammatory cytokine levels.Evaluate if interventions that reduce MCV or tackle root causes (such as nutritional deficiencies, oxidative stress) can prevent or alleviate liver fibrosis in high-risk groups.
Author Response
(Reviewer’s Comments)
- Investigate the biological pathways connecting MCV variations to liver fibrosis, including oxidative stress indicators, CHIP-associated genetic modifications, and inflammatory cytokine levels.
(Response)
The reviewer's suggestion to clarify the question in detail is very reasonable. However, since this study was a retrospective observational study based on electronic medical record data, detailed verification was not possible because samples could not be collected to explain the pathological mechanism. Since it is stated as a limitation that needs to be addressed in the future, we believe that readers will understand that it remains an unresolved issue.
- Evaluate if interventions that reduce MCV or tackle root causes (such as nutritional deficiencies, oxidative stress) can prevent or alleviate liver fibrosis in high-risk groups.
(Response)
As the reviewer pointed out, it is crucial to establish if normalizing the enlarged MCV with nutritional or pharmaceutical interventions that reduce oxidative stress can improve the severity of MASLD. In order to verify this issue obtained from the current observational study, it is necessary to conduct a prospective intervention study in the future. Regarding this matter, the following description has been added in the limitation. “Forth, it is unclear whether normalizing the enlarged MCV with nutritional or pharmaceutical interventions that reduce oxidative stress can improve the severity of MASLD.”
Round 2
Reviewer 2 Report
Comments and Suggestions for Authors
After the improvement, the article can be accepted.
Author Response
We would like to thank reviewer 2 for his/her appropriate additional comments. We have added the study limitations as suggested.